# Influence of Porosity on R-Curve Behaviour of Tetragonal Stabilized Zirconia

**Dino N. Boccaccini** [1,*][ID]**, Vanesa Gil** [2,3][ID]**, Jonas Gurauskis** [3,4][ID]**, Rosa I. Merino** [4]**, Andrea Pellacani** [5][ID]**, Cecilia Mortalò** [6][ID]**, Stefano Soprani** [7]**, Marcello Romagnoli** [5,8][ID] **and Maria Cannio** [1,*]

1    Resoh Solutions S.R.L., Via Pietro Giardini 476/N, 41124 Modena, Italy
2    Aragon Hydrogen Foundation, Parque Tecnológico Walqa, Ctra N330, km 566, 22197 Huesca, Spain
3    ARAID Foundation, Avda. Ranillas 1-D, 50018 Zaragoza, Spain
4    Instituto de Nanociencia y Materiales de Aragón, Universidad de Zaragoza-CSIC, 50018 Zaragoza, Spain
5    Dipartimento di Ingegneria Enzo Ferrari, Università degli Studi di Modena e Reggio Emilia, Via P. Vivarelli 10, 41125 Modena, Italy
6    Istituto di Chimica della Materia Condensata e di Tecnologie per l'Energia, Consiglio Nazionale delle Ricerche (CNR-ICMATE), Corso Stati Uniti 4, 35127 Padova, Italy
7    Carlsberg Research Laboratory, Brewing Science and Technology, J.C. Jacobsens Gade 4, 1778 Copenhagen, Denmark
8    H2—MO.RE, Centro Interdipartimentale di Ricerca e per i Servizi nel Settore della Produzione, Stoccaggio ed Utilizzo dell'Idrogeno, Via Università 4, 41121 Modena, Italy
*    Correspondence: dinoboccaccini@hotmail.com (D.N.B.); maria.cannio@gmail.com (M.C.)

**Abstract:** $Y_2O_3$ at 3% mol partially stabilized $Zr_2O_3$ (3YSZ) porous specimens with variable open porosity, from fully dense up to ~47%, and their potential use as anode supports for new solid oxide cell designs were fabricated by tape casting. The stiffness, strength and fracture properties were measured to investigate the influence of porosity on mechanical properties. The evolution of Young's modulus and characteristic strength was evaluated by ball-on-ring tests. The variation of critical plane stress Mode I stress intensity factor with porosity has also been investigated and modelled from the results obtained from fracture mechanics testing. R-curve behaviour was observed in dense 3YSZ specimens and in porous 3YSZ compositions. The width of the transformation zone after fracture mechanics testing and the variation with porosity were investigated. The phases existing in the fracture zone were determined and quantified by Raman spectroscopy. It was found that the width of the transformation zone increased with increasing porosity. A new general R-curve model for 3YSZ based on the McMeeking–Evans equation is presented, which can be used to predict the behaviour of the R-curve as a function of porosity, simply by knowing the properties of the dense material and introducing in this equation porosity-dependent laws on the key properties that affect fracture toughness.

**Keywords:** partially stabilized zirconia; transformation zone; Raman spectroscopy; R-curve behaviour

## 1. Introduction

Solid oxide cells are high temperature electrochemical devices that can either be operated as solid oxide fuel cells (SOFC) converting fuel to electricity or operated as solid oxide electrolysis cell (SOEC) converting electricity and $H_2O$ and/or $CO_2$ to fuel (hydrogen, CO or synthesis gas). In the current designs of solid oxide cell structures, the electrochemically active layers (anode, cathode) are processed to relatively thin geometries (~10 μm) to obtain a high power density at the working temperature [1]. However, these thin ceramic layers are intrinsically brittle and therefore should be integrated on top of a structural support layer, which is either an extension of the fuel electrode or the electrolyte [2]. A commonly used configuration for the manufacturing of SOFC and SOEC is the anode-supported or fuel electrode-supported configuration (e.g., Ni/YSZ [1] or Ni/GDC) [3]. Other alternative cell designs currently being developed are metal-supported cells [4], cathode

or oxygen electrode-supported cells [5] and electrolyte-supported cells [6]. In the LSM (oxygen electrode)-supported cell, the support is typically based on a thick LSM/YSZ layer placed on the oxygen side of the cell. This later design presents the drawback that during sintering above 1000 °C the reaction between LSM and YSZ into insulating zirconates may lead to a detrimental effect on the electrochemical performance [7]. To avoid this, new configurations based on a co-sintered porous support of YSZ into which a percolating electronic conducting phase is infiltrated after sintering have been studied [5]. It is worth mentioning that at such a high temperature, the so-called precipitation of impurities may occur, affecting the transformation of the crystal structure [8]. These backbones of porous 3YSZ must have a suitable porous microstructure to allow the infiltration to introduce sufficient electronic conductivity into the support structures [9]. Nevertheless, in addition, these porous backbones must integrate sufficient mechanical robustness to also serve their intended structural application [9]. Independent of the usage of porous 3YSZ—anode, cathode or other purposes, it is of high importance to establish porosity effect on structural performance of these porous backbones, especially bearing in mind the well-known effect of pore density and orientation on the reliability and mechanical properties of ceramic materials [10–12]. Apart from thermal expansion and compositional compatibility with the functional solid oxide cells (SOCs) layers, backbones based on 3% mol $Y_2O_3$ partially stabilized $Zr_2O_3$ (3YSZ) are characterized by the transformation toughening effect, which is a characteristic increase in fracture toughness arising in a material due to a phase transformation occurring at a tip of an advancing crack [13]. However, there are a few requirements that must be accomplished for a successful transformation toughening. First, there must be a metastable phase present in the material and the transformation of this phase to a more stable state must be capable of being stress-induced in the crack-tip stress field. Second, the transformation must be virtually instantaneous and not require time-dependent processes, such as long-range diffusion. Third, it must be associated with a shape and/or volume change. It is this last characteristic, the deviatory character of the transformation, which allows it to be stress-induced. It also provides the source of the toughening because the work done by the interaction of the crack-tip stresses and the transformation strains dissipates some of the energy that would normally be available for crack extension [13]. An alternative, but essentially equivalent, way of regarding the toughening process is as a form of crack shielding, in which the transformation strains generate local stresses that oppose further crack opening. It is well-known in the scientific community that transformation toughening occurring in dense partially stabilized zirconia (PSZ) can give rise to rising fracture resistance with crack extension (R-curve behaviour) [14–20] caused by the characteristic transformation toughening mechanism occurring in these materials. The transformation toughening effect is due to the fact that at sufficiently high stress, metastable particles undergo a transformation from the tetragonal to the monoclinic phase, which is accompanied by a volume increase of 4% [21]. Since the transformation is stress-induced, a zone of material containing transformed particles surrounds the crack tip after critical transformation stress has been exceeded [22]. Subcritical crack growth (SCCG) is another phenomenon observed in PSZ and often reported in the literature [19,23–27]. Generally, in truly brittle ceramic materials, crack growth occurs at a velocity that is in the order of magnitude of the speed of sound (fast fracture) after the applied energy release rate reaches a critical value ($G_{Ic}$). However, in some materials, including ceramics, crack growth can occur at lower velocities and this phenomenon is called subcritical crack growth (SCCG) [28–30]. SCCG can occur in reactive specimens due to the weakening of inter-atomic bonding at the crack tip by chemical interaction with the environment or by the transport of molecules to the crack front [27]. A higher concentration of water molecules in the environment increases the crack growth rate by favouring the cleavage of Zr–O–Zr bonds at the crack tip [23]. In this work, dense and porous $Y_2O_3$ partially stabilized $ZrO_2$ (3YSZ) materials were fabricated by tape-casting technique to prepare porous backbones for applications in the field of SOCs. The main purpose of this article is to provide perspective on porous structures with spherical pores and open porosity in the 1–47% porosity range prepared

by using polymethylmethacrylate (PMMA) as pore former. Porous structures with the highest pore volume-to-surface area ratio possible and with better mechanical properties (limited notch-features) were selected to investigate the influence of porosity on mechanical properties of 3YSZ. The tetragonal-monoclinic transition in 3YSZ was determined by micro-Raman spectroscopy [31,32]. The aim of the work was to investigate the influence of porosity on the mechanical properties of 3YSZ with different values of open porosity prepared by tape-casting. To this end the elastic modulus, flexural strength, Young's modulus and fracture toughness of 3YSZ supports with different open porosity (<1 to 47%) were deeply evaluated. The variation of transformation toughening effect as function of porosity and the influence of transformation zone on R-curve behaviour and the possible occurrence of SCCG was also investigated. Finally, a new model for R-curve behaviour based on McMeeking–Evans equation and including the influence of porosity on transformation toughening is proposed [14,33].

### 1.1. Theory

#### 1.1.1. Increase in Fracture Toughness from Transformation Toughening

As mentioned above, the rising fracture toughness ($K_R$) of zirconia ceramics is the result of an increasing process zone of stress induced martensitic transformation from tetragonal to the monoclinic structure at the crack tip region [9]. The stress-induced transformation occurring at the crack tip produces a transformation zone of height 2 $d$, where $d$ is the width of the transformation zone at one side of the advancing crack [34]. In most of the mechanistic models of transformation toughening, the formation of the initial transformation zone at the tip of a stationary crack has no net effect on the toughness of the material [13]. However, as the crack grows, material unloading occurs in the transformed material behind the crack tip. It is the non-reversible stress–strain behaviour (the stress–strain relation during unloading differs from the stress–strain relation during loading) that a material point undergoes around the crack tip, which leads to an increase in fracture toughness. The phenomenon of rising fracture toughness with increasing crack length is called R-curve behaviour [14–19]. Hence, the size of the transformation zone at the crack flanks is an important parameter, which depends on the temperature, the amount and type of stabilizer used, and the size of tetragonal zirconia particles [35].

The maximum capability of a material to experience transformation toughening can be expressed as a shielding stress intensity factor, $\Delta K_{R_{SS}}$, which is a function of the transformation zone shape, as evaluated by the parameter $\eta$, and the zone size ($d$). Thus, the increase in toughness from initiation $K_{R_0}$ to steady-state, $K_{R_{SS}}$, resulting from stress-activated transformation, $\Delta K_{R_{SS}}$, is commonly given by an expression of the form [14,33]:

$$\Delta K_{R_{SS}} = \frac{\eta \, E \, e_T V_f d^{1/2}}{1 - \nu} \tag{1}$$

where $\eta$ is a factor depending on the zone shape at the crack tip and the nature of the stress field in that zone, $E$ is the effective modulus of the material, $e_T$ is the dilatational strain, $V_f$ is the transformed volume fraction of particles, $d$ is the width of the transformation zone from the crack surface (i.e., the half-height of the zone), and $\nu$ is the Poisson ratio. Different values for $\eta$ have been proposed depending on the elongation of the zone ahead of the crack tip [14]. The value of $\Delta K_{R_{SS}}$ in Equation (1) indicates the maximum increase in the applied stress intensity factor $K$ that a material can withstand by transformation toughening mechanisms.

#### 1.1.2. R Curve Models

Evans has suggested an arctan function to fit the R-curve behaviour numerically calculated from the crack-shielding model [14,33]. The original equation, used to describe

the fracture toughness as a function of the crack length, is the McMeeking–Evans formulation [14,33]:

$$\Delta K(\Delta a) = \frac{\eta \, E \, e_T V_{fd}}{1 - v} \tan^{-1}\left(\frac{\Delta a}{a_0}\right) \tag{2}$$

where $V_{fd}$ the product between the transformed volume fraction of particles $V_f$ and the square root of the transformation zone width from the crack surface $d^{0.5}$ (i.e., the half-height of the zone), $\Delta a$ is the crack length and $a_0$ is a normalizing parameter inverse of the zone width ($d$) [14,33,36].

### 1.1.3. Our Model: Porosity-Dependent Model for R Curve Behaviour

With the aim to introduce a new generic model of R-curve behaviour, able to predict the performance of porous materials by just introducing the properties of the dense material and porosity-dependent laws for those properties affecting the R-curve behaviour, we have proposed the approach explained below. The dependence of the elastic modulus on the porosity has been introduced in the following terms [10,37,38]:

$$E(p) = Ee^{-b_M p} \tag{3}$$

where $E$ is the elastic modulus of the fully dense material, $b_M$ is a porosity dependence constant and $p$ is the porosity. The transformed volume fraction and width of the transformation zone are affected in two opposite ways by the porosity change. $V_f$ is expected to decrease linearly when the porosity grows; increasing the porosity of the material means to reduce the total full volume as well as the transformed volume. The linear influence of $V_f$ is attenuated by the change of $d$, though; the transformation zone width sensibly rises with porosity, as it is noticeable from the Scanning Probe Image (SPI) images obtained from nano-indentation testing and reported in the Results section below but also quantitatively, as determined by Raman spectroscopy. As the volume fraction of transforming particles decrease with the porosity, the overall expected trend for $V_{fd}$ is a function that decreases with porosity as follows:

$$V_{fd}(p) = V_{fd}(1 - p^s) \tag{4}$$

where $V_{fd}$ is the product between the transformed volume fraction of particles and the square root of the transformation zone width ($d$) of the full dense material, $p$ fractional porosity and $s$ a best fitting parameter, expected to be smaller than 1. Additionally, it can be expected that the width of the transformation zone decreases with the stiffness of the transforming zone, why we propose a similar porosity-dependent law was assumed for $d^{-1}$, the inverse of the transformation zone, as shown in Equation (5). The fracture toughness trend, following an arctangent according to McMeeking–Evans formulation for full dense materials, is also inquired through another fitting parameter $m$, exponent of the $\tan^{-1}$ term [14,33,36]. Finally, the resulting correlation proposed here, which expresses the fracture toughness as a function of the crack length, porosity and two best fitting parameters is the following one:

$$K(\Delta a, p, s, m) = \frac{\eta \left(Ee^{-b_M p}\right) e_T V_{fd}(1 - p^s)}{1 - v}\left(1 + \left(\tan^{-1}\left(\frac{\Delta a}{d\,(1 - p^s)}\right)\right)^m\right) \tag{5}$$

Therefore, by introducing the variation with porosity of the material properties in Equation (5) a general model of the materials behaviour can be obtained.

## 2. Materials and Methods

### 2.1. Materials

Porous 3 mol% yttria stabilized zirconia (3YSZ) (hereafter also called backbones) with different levels of open porosity ($P_0$) (<1–47%) and closed porosity ($P_c$) were prepared by tape casting. The porosity was modulated using PMMA as pore former at different

concentrations, being 0 wt%, 25 wt% and 50 wt% for BB05, BB06 and BB07 samples, respectively. Table 1 reports the pore characteristics, specifically open and closed porosities and average pore size based on volume, as derived from the mercury intrusion tests. The tape casting slurries where made using ethanol as solvent and contained PMMA as pore former (PMMA 7–10 μm from Esprix) and an in-house binder-dispersant system added to the 3YSZ powder (Tosoh Co.), as described in [1]. All the powders were used as delivered. The dried tape cast layers were laminated together to obtain thicker samples to achieve a better handling strength. Circular samples were punched out of the laminates to a diameter of ca. 25 mm and a thickness of ca. 300 μm and sintered at a temperature of 1315 °C (15 °C/h to 600 °C/4 h; 60 °C/h to 1315 °C/12 h; 100 °C/h to 25 °C end). For each type of backbone, between 29–40 samples were fabricated for ball-on-ring testing. Samples for other mechanical tests were made through the laser cutting of the sintered samples into the appropriate size (sample sizes are reported below).

**Table 1.** Porosity and average pore diameter measured by mercury intrusion porosimeter. The relative standard deviation (RSD) of porosity is 5%.

| Sample Name | Average Pore Diameter (μm) | Open Porosity (%) | Closed Porosity (%) |
|---|---|---|---|
| BB05 | - | 2.8 | - |
| BB06 | 0.32 | 13 | 11 |
| BB07 | 0.85 | 47 | 5 |

### 2.2. Microstructural Characterization

Recently, several methods of research and analyses of pores to investigate their dynamics and structure have been presented [39]. However, in this work, the porosity and pore size distribution of the porous samples were determined by standard mercury intrusion using an Autopore IV 9500V1.05 from Micromeritics Instrument Corporation, Norcross, GA. Due to the uncertainty of the mercury intrusion porosimeter for highly dense materials, the porosity of the dense sample was measured on an AccuPyc-1340 Helium Pycnometer. Scanning electron microscopy (Hitachi TM1000tabletopSEM) was performed on the sintered samples to investigate their microstructure. Prior to SEM investigation, the samples were vacuum embedded in Epofix (Struers, Denmark), ground and polished to 1 μm and coated with carbon to eliminate surface charging.

### 2.3. Nano-Indentation Testing

Nanoindentation experiments were performed on the smooth-cut surfaces of samples with a TI-950 TriboIndenter (Hysitron Inc., Minneapolis, MN, USA). The tip is a cone indenter with a nominal diameter of 5 μm. The machine compliance is 0.46 nm/mN. These indentation experiments were conducted in a quasistatic mode by applying the same load of 0.5 N in all samples. The tests were conducted under load control. The time for reaching the maximum load was 5 s, then followed by a hold of 2 s and unloading in 5 s. The in-situ SPM imaging mode was used to raster-scan the sample zone prior to the indentations and to identify the spots most suited for the indentation experiment (smoothness and level). The in situ SPM image was needed to identify the two distinct sample zones: the tetragonal phase base material and the area of the tetragonal to monoclinic transformation zone. The indenter used for the imaging procedure was a Cube Corner indenter geometry.

### 2.4. Raman Spectroscopy

Raman spectroscopy analyses were performed using either an optical micro-Raman spectrometer (Model XY, DILOR, Lille, France) or a confocal Raman equipment (Alfa 300 from WITec GmbH, Ulm, Germany). The method relies on the differences in the lattice phonon mode position and intensities of the corresponding Raman peaks for both tetragonal and monocline phases [32]. The existence of relatively small amounts of monoclinic phase is well established by the presence of the Raman double peak at 180 and 190 cm$^{-1}$ between the

150 and 260 cm$^{-1}$ tetragonal zirconia Raman peaks [32]. The ratio between the intensities of this double peak and that of the 150 cm$^{-1}$ Raman peak of tetragonal t-ZrO$_2$ can be used to determine the relative amount of monoclinic versus tetragonal phase. In the present case, the minimum monoclinic fraction that can be detected by this method is about 0.01. It is also interesting to note that Raman peak positions change with composition and stress. The measurements were taken along lines perpendicular to the fracture top surfaces. To obtain reference spectra of the non-deformed ceramic, Raman spectra were also taken perpendicular to the ceramic plate surface well far away from the fracture or notch surface. In the case of the micro-Raman set up, spectra were excited with the 514.5 mm line of an Ar$^+$ ion laser and collected at room temperature in a backscattering geometry using a triple 0.5 m monochromator with a CCD detector coupled to an optical microscope with a spectral resolution of 0.3 cm$^{-1}$. A backscattering micro-Raman configuration with the objective ×100 was also used. The measured volume in this setup is of about 2 μm diameter and a variable depth, depending on the sample transparency. Due to the scattering of the exciting and collected light beam, the measured volume is larger and deeper in the case of the dense and more transparent BB05 sample than in that of a porous, strongly scattering sample. In addition, we performed measurements in the confocal Raman set up to obtain a better spatial resolution. In this case, the measurement was performed with ×100 objective, and a 25 μm pinhole selected a region of about 2 μm thickness, in the direction parallel to the laser beam, around the focus position. An objective of ×50 that selected about 4 μm in depth around the focus was also used. In this case, the excitation source was the 532 nm green laser line.

Procedure

In order to select the signal coming from the near-to-surface volume, which is of critical importance when it is necessary to measure a sample volume while remaining as close as possible to the fracture surface volume, the procedure employed was as follows. The laser was focussed into the sample to obtain a maximum Raman signal. Then it displaced the focus upwards, away from the sample, until the signal intensity decreased to half. In this condition, the spectra were accumulated until a good signal-to-noise level was obtained. This corresponds to the spectra on the surface. Subsequently, spectra were also recorded shifting the sample upwards by 2 or 4 microns (as indicated in the corresponding spectra) so as to focus the exciting light and collecting objective inside the samples below the surface. This distance by which the sample has been shifted with respect to the surface at focus is used in the rest of the manuscript to identify the point of focus within the sample. The actual focussing position should be somewhat larger due to refraction. Raman spectra were measured in three different sample regions: (a) in the pristine untreated ceramic surface; (b) in the surface left by the notch consisting of a thin resolidified crystalline layer; and (c) in the fracture edge.

### 2.5. Flexural Strength by Ball-on-Ring

Flexural strength was determined by the ball-on-ring method by means of an Instron testing machine (Model 1362 graded to 88R1632). The displacement was measured using an LVDT (linear variable differential transformer, range ±50 mm, resolution ±2.5 μm) located in the Instron drive unit. The diameter of the supporting ring was 16 mm, while the diameter of the ball was 3.96 mm. The loading speed was 0.2 mm/min.

### 2.6. Fracture Toughness by Double Cantilever Beam (DCB)

Rectangular specimens of $(10 \times 60 \times 0.3)$ mm$^3$ with a central notch (25 mm long and 0.1 mm tall) along the biggest dimension where the tapes were laser cut. An additional micro-notch, needed to ensure that initial cracking takes place in the middle of the ceramic layer, was sawed at the laser-cut notch root with a steel blade. The experimental procedure for sample preparation is reported in reference [19]. The ceramic layer was placed inside the grooves of two steel beams and glued using Scotch-WeldTMDP460 from 3 M as two-

part epoxy adhesive to form the test specimen. The height, H, of the steel beams was 5.95 mm and the width, B, 4.85 mm. The DCB specimens were loaded with pure bending moments, using a special fixture that consists of grips that lies on a base fixture [19,26,40]. The DCB specimen loaded with pure bending moments is a steady-state specimen, as under constant moments, the energy release rate (*G*) is independent of the crack length ($\Delta a$), as seen from Equation (1) in reference [40]. The value of *G* was determined by the measurement of the applied moment. The fixture was mounted on the XYZ stage of an optical microscope (DeltaPix, camera Infinity X-32) and the crack growth was measured by translating the stage in the specimen (XY). The magnification used in the optical microscope was between 100–200 X. All the tests were conducted at ambient conditions. A detail description of the loading procedure for the determination of the R-curve behaviour can be found in references [9,19]. Briefly, the samples were loaded gently by applying pure bending moments by the double cantilever rig until a crack appeared at the root of the micro-notch. This crack propagated a length $\Delta a_1$ and then stopped due to the characteristic resistance to crack propagation offered by 3YSZ materials. The energy release rate applied for this initial crack growth or its associated fracture toughness value ($K_{R_0}$) was then determined. A subsequent increase in the applied moment trough the DCB rig (hence in the applied energy release rate or stress intensity factor) induced another crack of length $\Delta a_2$. At this point, the total length of the crack is $\Delta a = \Delta a_1 + \Delta a_2$. Again, the values of $K_R$ and $\Delta a$ were recorded. This procedure was subsequently applied up to a certain value of applied moment, and the crack propagated without stopping until the total sample breakdown. The recording and calculation of these values of applied moments and crack lengths allowed the full R-curve behaviour of the material to be obtained. Clearly, the value of applied stress intensity factor producing the catastrophic failure ($K_{R_{SS}}$) and the corresponding total crack length value $\Delta a$, which is called critical crack length, were also determined.

## 3. Results

### 3.1. Microstructural Characterization

A Scanning Electron Microscope (SEM) was used to investigate the microstructure of broken samples after the Double Cantilever Beam (DCB) tests in areas at a side of the catastrophic crack path. In particular, Figure 1a–f shows the SEM images of the selected samples corresponding (a,b) to BB05, (c,d) to BB06 and (e,f) to BB07. The three materials differ according to the percentage of open porosity, being 2.8%, 13% and 47% for BB05, BB06 and BB07, respectively, as consequence of the PMMA amount added during the sample preparation.

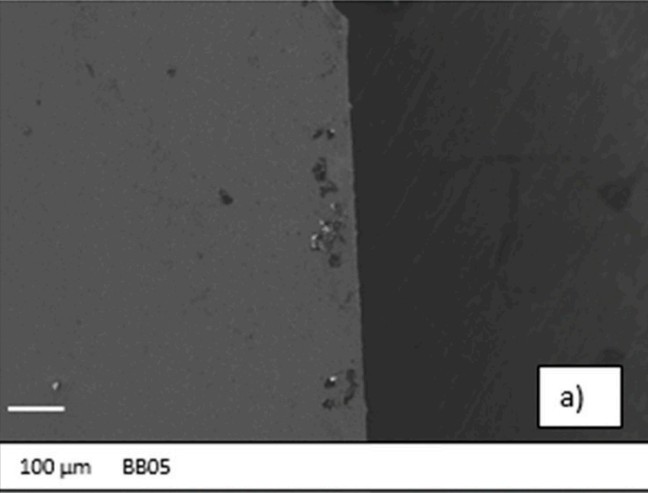
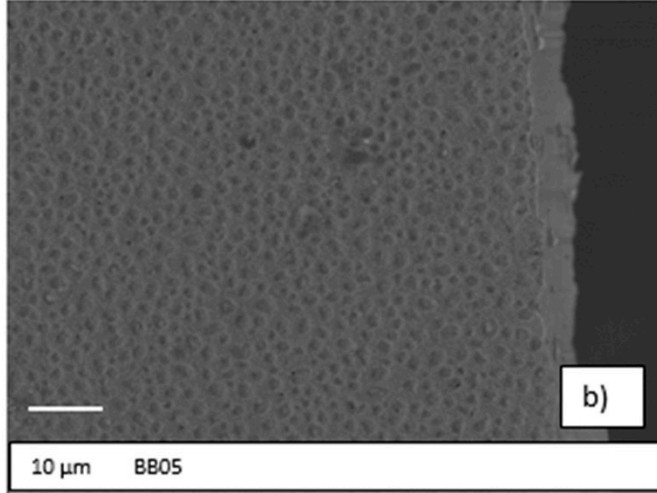

**Figure 1.** *Cont.*

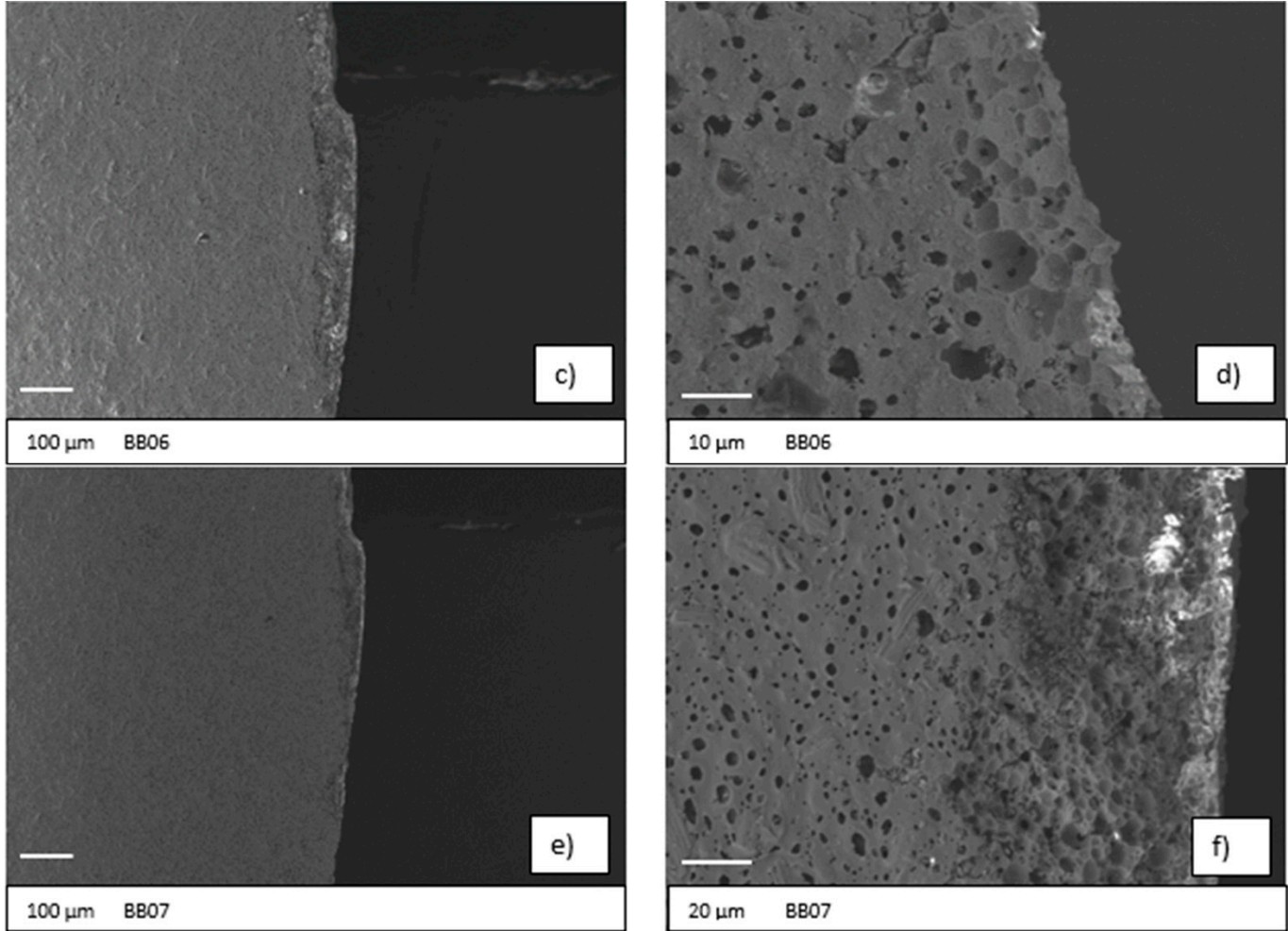

**Figure 1.** SEM images of the investigated samples.

*3.2. Biaxial Flexural Strength*

Table 2 summarizes the results for the elastic modulus and biaxial flexural strength on the prepared 3YSZ (BB) backbones measured at room temperature using the ball-on-ring method. The elastic modulus was calculated from the load deflection curves given by the ball-on-ring data. In reference [9], the experimental set-up used in the tests is described in detail. Furthermore, a systematic description of the influence of porosity on flexural strength of 3YSZ is also provided.

**Table 2.** Elastic and mechanical properties obtained by the ball-on ring tests of 3YSZ. Samples.

| Sample | Thickness (mm) | Number of Samples | $\sigma_0$ (MPa) | m | E (GPa) | $V_{eff}$ (mm$^3$) | $\sigma_{0,corr}$ ($V_{eff}$ = 1 mm$^3$) |
|---|---|---|---|---|---|---|---|
| BB05 | 0.3 ± 0.01 | 29 | 676 ± 55 | 5.5 | 214 ± 20 | 0.443 | 583 ± 47 |
| BB06 | 0.34 ± 0.01 | 31 | 264 ± 44 | 6.2 | 112 ± 18 | 0.3505 | 223 ± 37 |
| BB07 | 0.35 ± 0.01 | 40 | 143 ± 20 | 8 | 55 ± 8 | 0.214 | 118 ± 16 |

References [41,42] describe the procedure to calculate the Weibull characteristic strength ($\sigma_f$), Weibull modulus (m) and the effective volume ($V_{eff}$) for the different samples.

*3.3. Double Cantilever Beam: R-Curve Behaviour Determination in 3YSZ Bodies*

Figure 2 shows different images obtained during the double cantilever beam testing. In Figure 2a, it is possible to see the laser-cut notch. An additional micro-notch was sawed

at the laser-cut notch root to facilitate the crack onset and to ensure that the initial cracking takes place in the middle of the ceramic sample (Figure 2b). Figure 2c shows the onset of a crack after the initial loading propagating through the ceramic material starting from the notch and then stopping, as indicated by the black arrows. Figure 2d shows the already initiated crack propagating through the rest of the sample after an increase in the applied load. Figure 2d shows a complete picture of a representative sample (BB05) before and after the DCB testing and the path run by the propagating crack (Figure 2e,f).

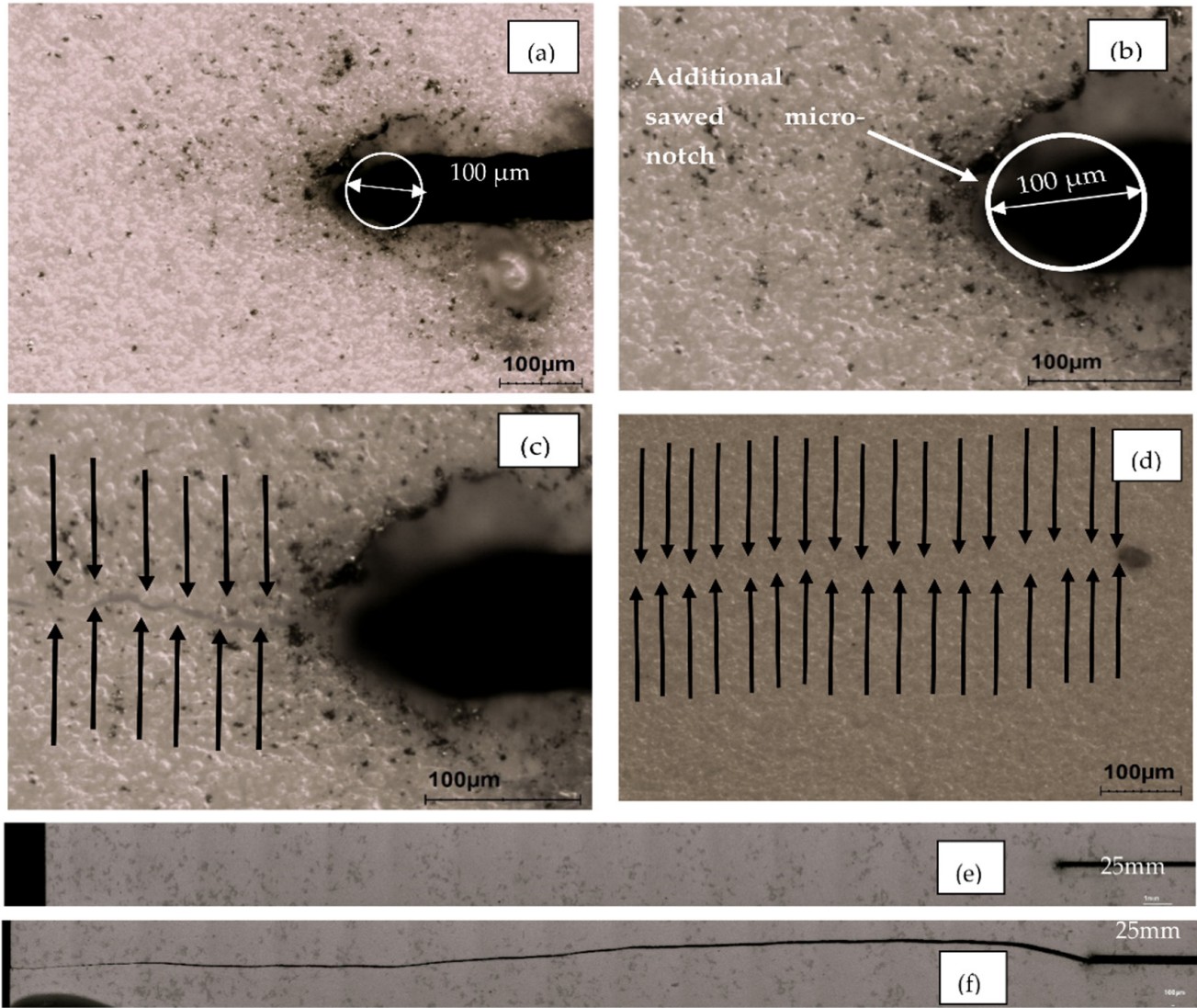

**Figure 2.** Images from double cantilever beam testing for a representative BB05 sample, (**a**) laser-cut notch and (**b**) additional sawed micro-notch, (**c**) arrows indicate the crack initiation, (**d**) arrows indicate the crack propagation and (**e**) sample before and (**f**) after catastrophic breakdown for $K_R \geq K_{R_{SS}}$.

### 3.4. R-Curve Behaviour

The double cantilever beam technique, following the settings and procedure described in the Experimental section and in references [19,26,43], was applied to determine the variation of fracture toughness with crack extension for dense and porous 3YSZ supports. Figure 3 plots the R-curve behaviours for BB05, BB06 and BB07 in terms of fracture toughness increase.

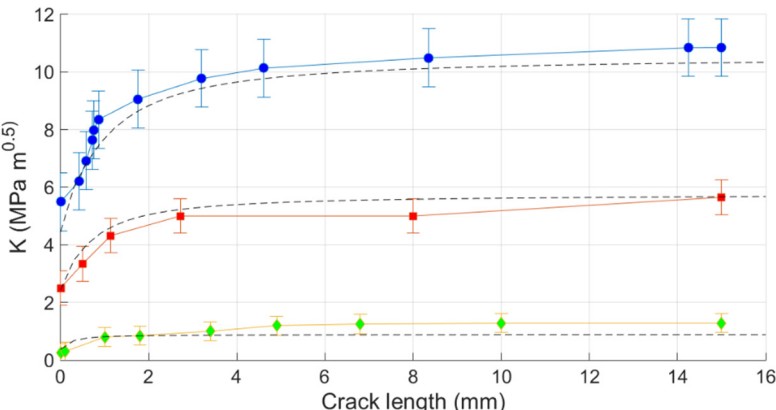

**Figure 3.** R-curve behaviour of 3YSZ samples with different porosity (*p*) values (standard deviation is also plotted). Dotted lines represent a family of curves of $K_I$ versus porosity obtained from Equation (5).

### 3.5. SPI Imaging: Nano-Indentation

Figure 4 shows the SPI images of the indentation tests for (a) dense (BB05) and (b,c) porous (BB07) samples.

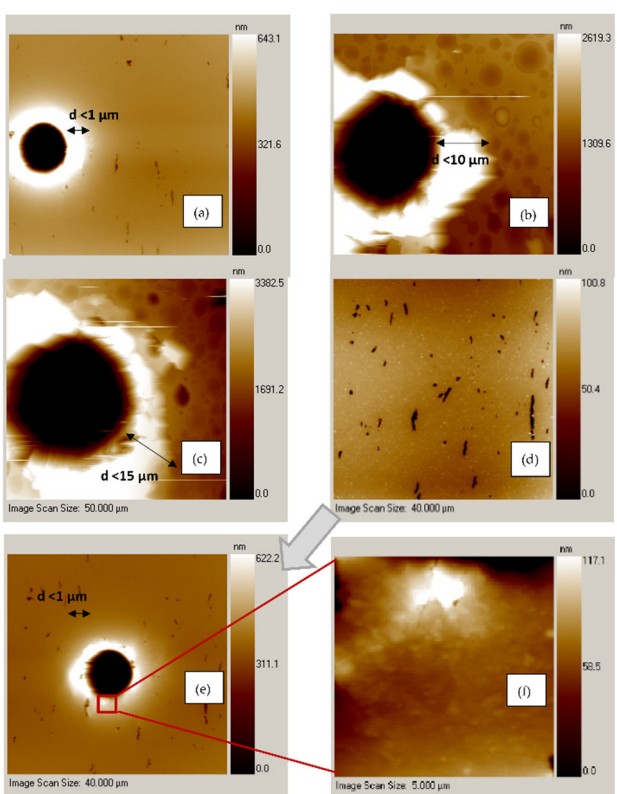

**Figure 4.** SPI images from indentation testing (spherical cone indenter of 5μm of diameter) for (**a**) BB05, (**b,c**) BB07 and (**d–f**) the enlargement of the microstructural damage due to indentation in a BB05 sample. (**d,e**) shows an area of a BB05 sample before and after the indentation, while (**f**) is an enlargement of the damaged area squared in red in (**e**).

### 3.6. Raman Spectroscopy

3.6.1. Raman Spectra Measured with the Micro Raman Equipment

Figure 5 shows the Raman spectra of pristine untreated and treated ceramic samples. The spectrum of the pristine untreated ceramic surface of a BB05 sample is observable in Figure 5, line a. It consists of five broad Raman peaks at 150, 260, 325, 465 and

640 cm$^{-1}$, which are characteristic of the t-ZrO$_2$ phase [44]. The faint structure between 370 and 425 cm$^{-1}$ is luminescence probably due to some impurities present in the material. Figure 5, line b) shows the Raman spectra measured with the micro-Raman setup in the fracture surface of the dense sample (BB05). The two spectra are essentially the same. In Figure 5, lines c,d the spectra corresponding to the samples with 13% (BB06) and 47% (BB07) porosity are displayed. In addition to the tetragonal peaks, two shoulders at 177 and 188 cm$^{-1}$ are seen (marked with arrows), which correspond to two of the 18 symmetry allowed Raman-active modes that can be predicted for the monoclinic ZrO$_2$ phase [44]. In samples with low m-ZrO$_2$ content these are the only well-observed Raman peaks as they are the narrower and more intense modes of this structure. In accordance with the method reported in reference [45], this Raman signal is ascribable to presence of about 1 vol% of m-ZrO$_2$ in the explored volume.

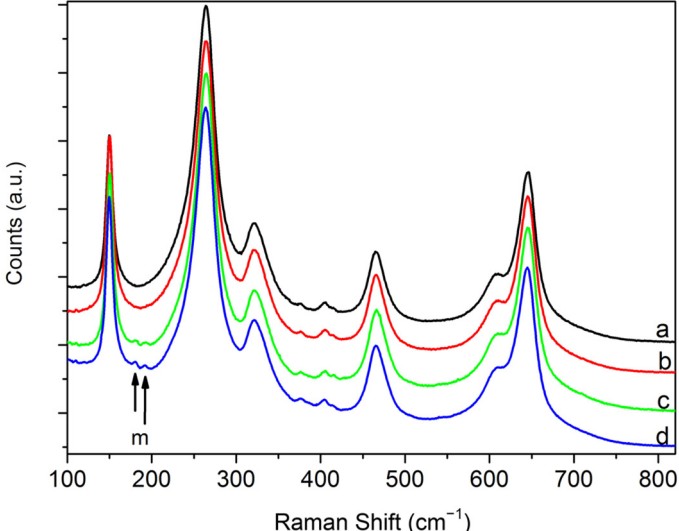

**Figure 5.** Raman spectra measured with the micro-Raman equipment (as described in the text). Ceramic surface (a); on fracture in dense sample BB05 (b); on a fracture in 13% porous ceramic BB06 (c); on a fracture in 47% porous ceramic BB07 (d). The arrows mark the bands that identify the monoclinic phase.

As a summary, micro-Raman technique detects the presence of monoclinic zirconia near the fractured surface in the porous samples, but no monoclinic zirconia is evident in the dense one.

3.6.2. Raman Spectra Taken with Confocal Raman Set Up

Figure 6 shows spectra measured with the confocal Raman in the fracture surface of the dense sample (BB05) and focussing on the surface and, nominally, at 2 and 4 microns below (sample stage shift). m-ZrO$_2$ is only detected clearly in the sample region at less than 2 microns of the fracture surface with a concentration of about 4.7 vol% in the explored volume.

Figure 7 shows the Raman spectra of the sample with 47% porosity (BB07). In contrast with the behaviour observed in the dense sample, here m-ZrO$_2$, is detected in all the spectra taken in the surface and focusing nominally (sample stage shift) at 2 microns and at 4 microns deep below the surface. On the other hand, these spectra did not show variation in the amount of m-ZrO$_2$ detected changing the depth of focus. Moreover, only a tiny amount, almost indistinguishable with the noise, was observed in all the spectra. In order to detect monoclinic zirconia, we had to use the x50 objective that explores a sample volume of about 4 microns depth around the focus position. Using the method outlined above the amount of m-ZrO$_2$ in this sample was estimated 2.9% with the focus on the surface, 3.6% at 2 and 4 microns from surface.

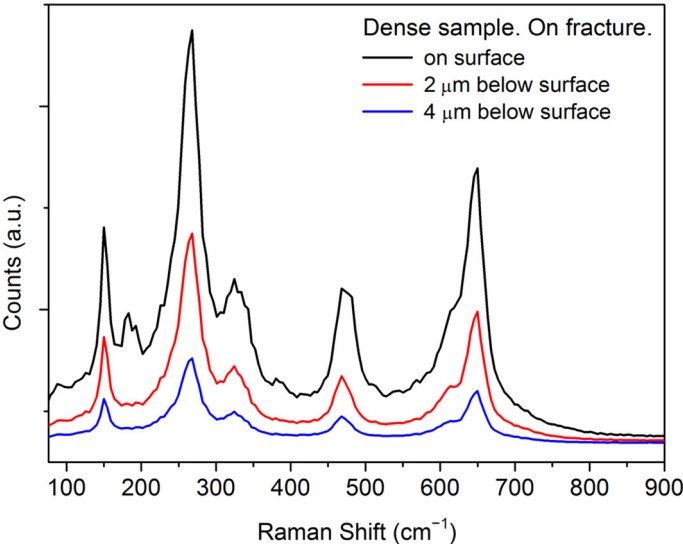

**Figure 6.** Raman spectra taken with confocal Raman set up on a dense sample. Black line in the surface; red line below surface, stage shifted 2 microns; blue line below surface, stage shifted 4 microns. Measurements with ×100, 0.9 NA objective.

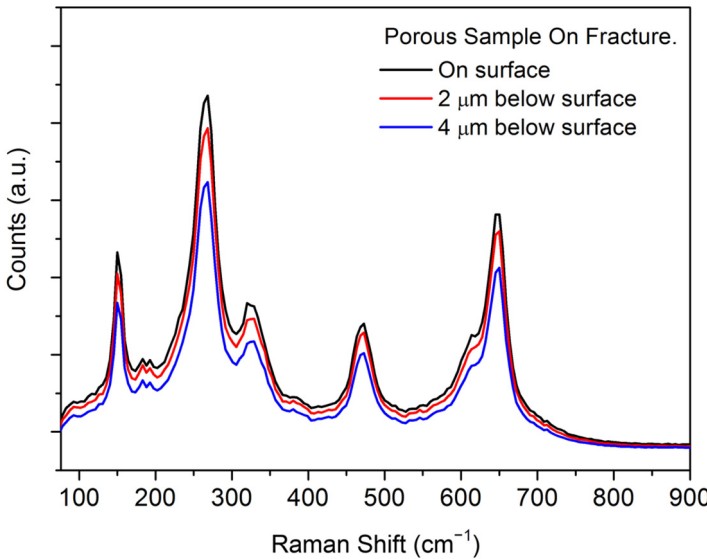

**Figure 7.** Raman spectra of a 47% porous sample, obtained with a confocal apparatus with ×50, 0.8 NA objective. At the surface (black line) and stage shifted 2 microns (red line) and 4 microns depth (blue line) into the sample.

### 4. Discussion

The SEM images shown in Figure 1 indicates that sample BB05 is well structured and homogeneous, while samples BB06 and BB07 are characterized by more discontinuities since a fraction of their volume is occupied by empty spaces of different shapes and sizes. Furthermore, it is possible to observe microstructural damage at the crack flanks. This damage could be ascribed to the typical crystallographic transformation from tetragonal to monoclinic phase. However, to validate and quantify this damage, a specific technique as Raman microscopy was performed. Figure 1a, image corresponding to the dense sample BB05, shows that the transformation width is very narrow, even at higher magnification (Figure 1b). In sample BB06, it is possible to distinguish a slightly more marked damaged area probably justified by the occurring of crystallographic changes. In sample BB07, the microstructural damage is well distinguishable even at reduced magnification compared

to previous ones. The PMMA was used as pore-former after sintering to obtain spherical pores of various sizes differently distributed in the sample volume. Regarding the results of the mercury intrusion tests reported in Table 1, sample BB07 shows the highest percentage of porosity and compared to sample BB06; it shows an increase of 34% in open porosity by adding 25 wt% more of the pore-former. If compared BB07 to the BB05 sample (with the amount of pore-former at 50 wt% and 0 wt%, respectively), the difference in open porosity achieved was 44.2%. These data indicate that the trend of the increase in porosity as function of pore-former content is not linear but could be better approached by an exponential growth. In addition, an increase in the average pore size as the porosity increases can be observed. This behaviour could be explained assuming that different PMMA concentrations favoured, in areas with a high content of pore-former, a physical interaction between the pores: an interaction/fusion of these pores resulted in a single pore of larger size. Table 2 reports the summary of the mechanical properties of the investigated samples. The data suggest a clear decrease in characteristic strength of 3YSZ with increasing porosity as expected [24,25,41,46,47]. In fact, the maximum characteristic strength value of 676 MPa was found for the denser sample (BB05), while for the sample with a higher porosity (BB07), equal to 47% porosity, the characteristic strength value was 143 MPa. Figure 2 shows different stages of the crack propagation through a representative BB05 sample. Figure 2a illustrate the dimensions of the laser made notch, while Figure 2b displays the additional micro-notch sawed to ensure that initial cracking take place in the middle of the ceramic layer. Figure 2c shows a crack that had "popped in" to the material (when $K_R$ is higher than $K_{R_0}$) from the notch due to the increasing pure bending moments applied by means of the DCB rig. This crack propagated through the sample until the increasing resistance offered by the material to the crack propagation stopped it. Subsequent increases in the applied moments allowed the crack to propagate through the sample, as Figure 2c shows. Then, when the applied $K_R$ was $\geq K_{R_{SS}}$, a catastrophic crack propagation occurred, leading to the total breakdown of the sample, as Figure 2e depicts. Figure 3 suggests that this phenomenon may occur once the total crack extension reached a value in the order of 10–12 mm for the dense BB05 sample, as the R-curve finds its own asymptote, meaning that the material does not offer any more resistance to the crack propagation. Figure 3 demonstrates that R-curve behaviour was observed in the dense 3YSZ specimens (BB05) but also in the porous BB06 (13% porosity) and BB07 (46% p.) samples. It is worth noting that in the 3YSZ backbones prepared in a previous investigation [9], with approximately the same porosity level as the BB07 samples in the present work, subcritical crack growth behaviour was observed instead of R-curve behaviour. This level of porosity (~50%) seems to apparently be a particular transitional zone in which 3YSZ porous backbones may show R-curve behaviour or subcritical crack-growth, depending on the raw materials, the procedure carried out during the sample preparation and the sintering conditions. Further investigations should be performed for a deeper understanding of the mechanical behaviour of 3YSZ for this particular porosity range. It is evident that the dense sample BB05 shows the higher initial fracture toughness ($K_{R_0}$) and this value strongly depends on the total porosity. Furthermore, the $\Delta K(\Delta a)$ decreases with the porosity content. For this reason, the steady-state $K_{R_{SS}}$ values, which are the highest fracture toughness that the materials can offer to contrast the crack propagation, also decrease with increasing porosity. The SPI images of Figure 4 demonstrate the development of transformation-induced uplift around the indents, which is in agreement with results reported in reference [48]. An important observation is that for the same applied load, these transformation-induced uplifts around the indents seems to increase for increasing porosities. This was later quantitatively assessed by the Raman spectroscopy analysis, the results of which are discussed below. Figure 5 shows the Raman spectra of pristine untreated and treated ceramic samples obtained with Micro Raman equipment. Micro Raman technique detects the presence of monoclinic zirconia near the fractured surface in the porous samples, but no monoclinic zirconia is evident in the dense one. Regarding the Raman spectra taken with confocal Raman set up, m-$ZrO_2$ is only detected in the sample region at less than 2 microns

from the fracture surface with a concentration of about 4.7 vol% in BB05 sample. In the case of the porous BB07 sample, m-ZrO$_2$ was estimated 2.9% with the focus on the surface, 3.6% at 2 microns from the surface and 3.6% at 4 microns. Interestingly, it has to be noted that the porous sample has less of the m-ZrO$_2$ phase than the dense sample and it is placed at about 2–4 microns below the fracture surface, i.e., for increasing porosity a less intense but a more widespread transformation zone is obtained. Summarizing, in the untreated regions, m-ZrO$_2$ was never observed, which demonstrates that the original ceramics is pure YTZ. m-ZrO$_2$ was observed in the fractured porous samples to a depth of about 2–4 microns. m-ZrO$_2$ was also observed in the dense samples but to a depth less than 1 micron from the fracture surface. In addition (results not shown here), in the melted surface of the laser made notch, m-ZrO$_2$ was always observed in the porous samples. This monoclinic zirconia may have its origin in cracking, resulting from the fast solidification of the debris produced by laser treatment. All in all, the results reported above obtained from different experimental techniques, e.g., nano-indentation and Raman microscopy, demonstrate the strong influence of porosity on the width of the transformation zone from both qualitative and quantitative points of view and, as a consequence, on the mechanical properties of the 3YSZ. It is clear that as the porosity increases, this transformation zone increases as well. This can be explained considering that transformation takes place in the grains in the stress field close to the fracture surface. In dense materials there is a higher number of grains per volume unit and, taking into account that the phase transformation takes place in the grains in the stress field close to the fracture surface, the presence of a higher grain density at the crack flanks can contribute to better dissipate the stresses and, as a consequence, the transformation width will be lower. After experimentally evaluating the influence of the porosity on the toughness of 3YSZ, some empirical correlations from the literature were used to model the observed behaviour. Correlations for toughness calculations do not take in account the porosity of the material, however. Therefore, some new porosity-dependent terms have been introduced in order to best fit the experimental data and generate a new, porosity-dependent, model for the 3YSZ R-curve behaviour. Equation (5) has been used in the best fit of the experimental data (Figure 3). It has been implemented in MATLAB and optimized according to the least square method. For the determination of $V_{fd}$, a value of 4.7% for $V_f$ and a $d$ value of 1.35 µm was assumed based on the results of the Raman spectroscopy. The parameters, used in Equation (5) and listed in Table 3, were taken from references [9,21,38] and from the experimental data obtained in this paper ($V_{fh}$, d).

**Table 3.** Parameters used for modelling the R-curve behaviour by Equation (5) taken from [9,21,38] and experimental data.

| Parameter | | Value | Unit | Ref. |
|---|---|---|---|---|
| $\eta$ | a factor depending on the zone shape at the crack tip and the nature of the stress field in that zone | 0.21 | / | [21] |
| $E$ | effective modulus of the material | 220,000 | MPa | [21] |
| $b_M$ | porosity dependence constant | 2.3 | / | [38] |
| $e_T$ | dilatational strain | 0.05 | / | [21] |
| $V_{fh}$ | transformed volume fraction of particles | 0.0015 | m$^{0.5}$ | This paper |
| $\nu$ | Poisson's ratio | 0.3 | / | [9] |
| $d$ | width of the transformation zone from the crack surface | $1.35 \times 10^{-6}$ | M | This paper |

Figure 3 shows the variation of the experimental data and the theoretical models obtained through Equation (5) for the obtained best-fitting parameters (m = 0.7 and s = 2.2). It is evident that the obtained theoretical models present a very good fit with respect to the experimental data: in fact, they fall within the margin of error of the experimental data methods employed represented by the standard deviation. A Student's *t*-test has also been carried out to validate these theoretical models. The *p* values obtained are 0.01, 0.04 and 0.05 for BB05, BB06 and BB07, respectively. These low values confirm the validity of

this approach and its effectiveness in predicting the influence of porosity on the fracture toughness of zirconia materials.

## 5. Conclusions

In this paper, the influence of porosity on the fracture toughness of 3YSZ was investigated and modelled. Vibrational Raman spectra on fracture surface of dense and porous samples were measured using two different techniques: micro-Raman and confocal Raman techniques. Both techniques are complementary with the confocal Raman suitable for dense and translucent samples, whereas in highly optically-scattering porous samples micro-Raman is more appropriated to obtain spectra from regions below the surface. Martensitic transformation from tetragonal to monoclinic YSZ, induced by mechanical stresses in the fracture region, was detected in all the YSZ-prepared samples. Interestingly, the transformed region is shallower in the dense sample, less than 2 μm thickness, being deeper in the porous regions, probably because the transformation takes place in the grains in the stress field close to the fracture surface. The latter can be deeper in the porous ceramics. Based on the results obtained by the Raman spectroscopy it was possible to model the influence of porosity on fracture toughness through the implementation of porosity depending on properties into the well-known McMeeking–Evans equation for fully dense materials. The goodness of fit of these models was demonstrated by statistical analysis (Student's *t*-test). The importance of this general model is that the R curves of porous samples can be completely described from theoretical models by just using the experimental data of the dense material.

**Author Contributions:** Conceptualization, D.N.B., S.S. and V.G.; methodology, D.N.B.; investigation, D.N.B., R.I.M. and V.G.; resources, J.G. and A.P.; data curation, D.N.B., M.C. and R.I.M.; writing— original draft preparation, D.N.B. and M.C.; writing— review and editing, D.N.B., M.C., M.R. and C.M.; visualization, M.R.; supervision, D.N.B., S.S. and C.M.; project administration, M.R.; funding acquisition, D.N.B. All authors have read and agreed to the published version of the manuscript.

**Funding:** This research received no external funding.

**Institutional Review Board Statement:** Not applicable.

**Informed Consent Statement:** Not applicable.

**Data Availability Statement:** Not applicable.

**Acknowledgments:** D.N.B. would like to thank Ude Hangen and Jaroslav Lukes for their help to perform the nano-indentation testing. Special acknowledgments to H.L. Frandsen for helpful discussion. R.I.M. would like to acknowledge V.M. Orera for his contribution.

**Conflicts of Interest:** The authors declare no conflict of interest.

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
