# Peer review of "Influence of Porosity on R-Curve Behaviour of Tetragonal Stabilized Zirconia"

_ceramics, doi:10.3390/ceramics5030040_

Round 1

Reviewer 1 Report

1. In the SEM, the BB06 seems to have large pore size than BB07, which is different in table 1, please check. 

2. How can you define the presence of about 2 vol% of m-ZrO2 in the explored volume by Raman? The question to Fig 6, 6.5% on the surface, 9.2% at 2 microns, and 9.5 at 4 microns from surface.

3. Why the porosity can have an effect on the t-ZrO2 to m-ZrO2 phase transition?

Author Response

Dear Reviewer,

please find my answers to the questions below.

1. In the SEM, the BB06 seems to have large pore size than BB07, which is different in table 1, please check. 

We ask the reviewer to consider that Figure 1 f) was performed at lower magnification than Figure 1f).

  1. How can you define the presence of about 2 vol% of m-ZrO2 in the explored volume by Raman? The question to Fig 6, 6.5% on the surface, 9.2% at 2 microns, and 9.5 at 4 microns from surface.

The amount of m-ZrO2 has been calculated from the intensity at the maximum of the Raman lines at 177 and 188 cm-1, which correspond to m-ZrO2 and the one at 150 cm-1, which corresponds to t-ZrO2. The actual calculation has been made as explained in the works of Baudín et al., J. Am. Ceram. Soc 92 (2009), 152-160 and Kim et al., J. Mat. Sci. Lett 16 (1997) 669-671. That is, the intensity ratios (Xm, as named by Kim et al.) wre calculated from the intesity at the maximum of each band, after a linear baseline crossing the background level at aproximately 135 and 205 cm-1 has been substracted. Then, the calibration curve given by Kim et al has been used to calculate the volume fraction of m-ZrO2 in each case.

The reference of Kim et al. is introduced in page 3 in the manuscript, dropped in the previous version by mistake. The same reference is included for clariy also in section 3.5.1.

Upon revising the text, we noticed a mistake in the transformation from intensity ratios to vol %, which has been corrected in the revised version at all places. The numbers are smaller. Nevertheless, the trends in the m-ZrO2 content and distribution in the samples does not change. That is, the manuscript results remain true.

  1. Why the porosity can have an effect on the t-ZrO2 to m-ZrO2 phase transition?

We thank the reviewer for rising this interesting point. An explanation for this it is provided in lines 588-591:  “It is clear that the higher is the porosity, this transformation zone increases. This can be explained considering that transformation takes place in the grains in the stress field close to the fracture surface. The later can be deeper in the porous ceramics”. However, we agree with the reviewer that it is too brief.

Therefore, we have added some more sentences to better clarify the influence of porosity on phase transition during crack propagation:

“In dense materials there is a higher amount of grains per volume unit and taking into account that the phase transformation takes place in the grains in the stress field close to the fracture surface, the presence of a higher grain density at the crack flanks can con-tribute to better dissipate the stresses and as a consequence the transformation width will be lower”.

You find in attachment the marked version of our manuscript.

Thank you very much for the useful suggestions., which help has to signficantly improve the quelity of our paper.

Best regards,

Maria Cannio

Reviewer 2 Report

Comments

This work presents an interesting combination of a large amount of data indicating the influence of porosity on R-curve behavior of tetragonal stabilized zirconia. Nevertheless, a rearrange of framework of this work can improve the manuscript, which will become easier and more approachable for the reader. More details are listed below:

1.      Please check all the explanations of abbreviations e.g: PMMA

2.      Pages 110-123 (spherical…stress) should be included in  another section, Materials and methods or discussion

3.      Lines 177-211 are better to go to the results section

4.      Before section 3 add the title Results

5.      Are there any EDS analyses indicating composition of the materials  or impurirties  e.g. mapping? If yes, please add.

  6. Can the impurities contained in micro areas of the material or in the lattice affect the  transformation of crystal structure and  finally the porosity?    

Good luck 

Kind regards

Author Response

Dear Reviewer,

please find our answers to your questions below.

1. Please check all the explanations of abbreviations e.g: PMMA

Done.

We added polymethylmethacrylate for PMMA

  1. Pages 110-123 (spherical…stress) should be included in another section, Materials and methods or discussion

Done. We moved some lines to the “Materials and methods” section 2.4, following the reviewer suggestion.

  1. Lines 177-211 are better to go to the results section

The authors think that this section must be included in the current theoretical section where our model is presented.

  1. Before section 3 add the title Results

 Done.

  1. Are there any EDS analyses indicating composition of the materials  or impurirties  e.g. mapping? If yes, please add.

We have not performed any EDS analyses, since we think it may be out of the topic of the work, which is focused on the impact of porosity on mechanical properties. We thank the reviewer for this indication; we will consider this for another work addressed to microstructural issues affecting the phase transformation.

  1. Can the impurities contained in micro areas of the material or in the lattice affect the transformation of crystal structure and finally the porosity?    

It is an interesting observation. The sample preparation was performed at DTU labs, especially dedicated to SOFC manufacturing. The presence of impurities was not considered based on the huge experience and severe control of DTU technical personnel performed during the application of the  process protocol. We agree with the reviewer that the presence of impurities contained in micro areas of the material or in the lattice may affect the transformation of crystal structure.

We have a new sentence and reference to clarify this point in line 56:

It is worth to mention that at such a high temperature, the so-called precipitation of impurities may occur, affecting the transformation of the crystal structure [Savoini, B.; Ballesteros, C.; Santiuste, J.M.; González, R.; Popov, A.I.; Chen, Y. Copper and iron precipitates in thermochemically reduced yttria-stabilized zirconia crystals. Philos. Mag. Lett. 2001, 81, 555–561]”

You find in attachment the marked version of our manuscript.

Thank you very much for the useful suggestions., which help has to signficantly improve the quelity of our paper.

Best regards,

Maria Cannio

Reviewer 3 Report

This is a fairly solid work, which should undoubtedly be recommended for publication, but after clarifying some incomprehensible points.

1.     Line 40.  3YSZ abbreviation needs to be deciphered.

2.     Line 52. The end of the sentence needs supporting reference.

3.     Line 53. Note that at such a high temperature, the so-called precipitation of impurities occurs:

Savoini, B.; Ballesteros, C.; Santiuste, J.M.; González, R.; Popov, A.I.; Chen, Y. Copper and iron precipitates in thermochemically reduced yttria-stabilized zirconia crystals. Philos. Mag. Lett. 200181, 555–561

Thus, requirements and precise dopant concentrations are important.

4.     Line 59. The end of the sentences needs supporting reference.

5.     Line 69-77.  The same.

6.     Lines 105 -113. More information about the methods of research and analysis of pores, their dynamics and structures would be extremely useful, especially for attracting a wide range of readers. See, recent MDPI paper and references therein: Klym, H., et al (2021). Positron annihilation lifetime spectroscopy insight on free volume conversion of nanostructured MgAl2O4 ceramics. Nanomaterials11(12), 3373.

7.     Line 50. “perhaps cite Khajavi” ???

8.     In formulas (1-3, 5), it is not always clear which indices are which.

9.     Were there any effects of aging, according to the SEM pictures taken at different times?

10.  In the table 2, instead of Gpa, there should be GPa.

11.  The quality of Figure 3 is unsatisfactory, not all details are visible.

12.  Line 464. instead of 4microns, there should be 4 microns.

13.  Table 3. Deciphering the symbols (x, E, b etc ...) would be useful.

14.  Correct reference 2.

Author Response

Dear Reviewer,

please find our answers to your questions below.

  1. Line 40.  3YSZ abbreviation needs to be deciphered.

Done. We have deciphered the abbreviation the first time it is used (in the abstract, line 20)

  1. Line 52. The end of the sentence needs supporting reference.

Line 52 has already a supporting reference (number 7):

However, we added a new reference at the end of line 58, maybe the reviewer has a different Line numbering.

  1. Line 53. Note that at such a high temperature, the so-called precipitation of impurities occurs:

Savoini, B.; Ballesteros, C.; Santiuste, J.M.; González, R.; Popov, A.I.; Chen, Y. Copper and iron precipitates in thermochemically reduced yttria-stabilized zirconia crystals. Philos. Mag. Lett. 200181, 555–561

Thus, requirements and precise dopant concentrations are important.

Thanks for this suggestion. We have included this reference and the following sentence to clarify the influence of the impurities on crystallization:

It is worth to mention that at such a high temperature, the so-called precipitation of impurities may occur, affecting the transformation of the crystal structure [Savoini, B.; Ballesteros, C.; Santiuste, J.M.; González, R.; Popov, A.I.; Chen, Y. Copper and iron precipitates in thermochemically reduced yttria-stabilized zirconia crystals. Philos. Mag. Lett. 2001, 81, 555–561]”

The sample preparation was performed at DTU labs, especially dedicated to SOFC manufacturing. The presence of impurities was not considered based on the huge experience and severe control of DTU technical personnel performed during the application of the process protocol.

  1. Line 59. The end of the sentences needs supporting reference.

Unfortunately, the reviewer Line numbering is different from our manuscript file downloaded from the journal web site. We have added some more references where we think the reviewer was suggesting.

  1. Line 69-77.  The same.

Unfortunately, the reviewer Line numbering is different from our manuscript file downloaded from the journal web site. We have added some more references where we think the reviewer was suggesting.

  1. Lines 105 -113. More information about the methods of research and analysis of pores, their dynamics and structures would be extremely useful, especially for attracting a wide range of readers. See, recent MDPI paper and references therein: Klym, H., et al (2021). Positron annihilation lifetime spectroscopy insight on free volume conversion of nanostructured MgAl2O4 ceramics. Nanomaterials11(12), 3373.

Following the reviewer indication, we have added this paragraph and reference at the beginning of the section 2.2

“2.2 Microstructural characterization

Recently several methods of research and analysis of pores to investigate their dy-namics and structure has been presented [Klym, H., et al (2021). Positron annihilation lifetime spectroscopy insight on free volume conversion of nanostructured MgAl2O4 ceramics. Nanomaterials, 11(12), 3373.]. However, in this work, the porosity and pore size distribution of the porous samples were determined by standard mercury intrusion using an Autopore IV 9500V1.05”

  1. Line 50. “perhaps cite Khajavi” ???

We have added this reference in Line 149.

  1. In formulas (1-3, 5), it is not always clear which indices are which.

We have checked carefully all the equations and all the parameters present in the equations are defined in the text.  

  1. Were there any effects of aging, according to the SEM pictures taken at different times?
  2. In the table 2, instead of Gpa, there should be GPa.

Done

  1. The quality of Figure 3 is unsatisfactory, not all details are visible.

Thanks for the suggestion; we have improved the quality of the figure.

  1. Line 464. instead of 4microns, there should be 4 microns.

Done.

  1. Table 3. Deciphering the symbols (x, E, b etc ...) would be useful.

Done.

  1. Correct reference 2.

Done

You find in attachment the marked version of our manuscript.

Thank you very much for the useful suggestions., which help has to signficantly improve the quelity of our paper.

Best regards,

Maria Cannio

Round 2

Reviewer 3 Report

The authors significantly improved their manuscript, which now can be recommended for publication